# Dynamics Learning Rate Bias in Pigeons: Insights from Reinforcement Learning and Neural Correlates

**DOI:** 10.3390/ani14030489

**Published:** 2024-02-01

**Authors:** Fuli Jin, Lifang Yang, Long Yang, Jiajia Li, Mengmeng Li, Zhigang Shang

**Affiliations:** 1School of Electrical and Information Engineering, Zhengzhou University, Zhengzhou 450001, China; jinfuli0313@126.com (F.J.); flyer1014@163.com (L.Y.); longyang_zzu@163.com (L.Y.); jiajia_li@gs.zzu.edu.cn (J.L.); 2Henan Key Laboratory of Brain Science and Brain-Computer Interface Technology, Zhengzhou 450001, China; 3Institute of Medical Engineering Technology and Data Mining, Zhengzhou University, Zhengzhou 450001, China

**Keywords:** asymmetric learning rate, dynamic learning rate bias, striatum, value, local field potentials, pigeon

## Abstract

**Simple Summary:**

Our study investigated how pigeons learn and make decisions. We found that pigeons showed a tendency to learn differently from good and bad experiences similar to humans and other animals; a pattern we call learning rate bias. Initially, pigeons are cautious and tend to learn more from the negative outcomes. As they learn and gain experience, they start to become more optimistic, favoring learning from the positive outcomes. We observed these changes through behavioral models and the neural activity in the striatum, a brain region involved in reward and learning. The shift from a cautious to a more confident approach in their decision-making suggests that pigeons use a flexible strategy for learning, which changes as they gather more information about what to expect from their choices. Understanding these patterns can help us learn more about basic learning processes shared across different species.

**Abstract:**

Research in reinforcement learning indicates that animals respond differently to positive and negative reward prediction errors, which can be calculated by assuming learning rate bias. Many studies have shown that humans and other animals have learning rate bias during learning, but it is unclear whether and how the bias changes throughout the entire learning process. Here, we recorded the behavior data and the local field potentials (LFPs) in the striatum of five pigeons performing a probabilistic learning task. Reinforcement learning models with and without learning rate biases were used to dynamically fit the pigeons’ choice behavior and estimate the option values. Furthemore, the correlation between the striatal LFPs power and the model-estimated option values was explored. We found that the pigeons’ learning rate bias shifted from negative to positive during the learning process, and the striatal Gamma (31 to 80 Hz) power correlated with the option values modulated by dynamic learning rate bias. In conclusion, our results support the hypothesis that pigeons employ a dynamic learning strategy in the learning process from both behavioral and neural aspects, providing valuable insights into reinforcement learning mechanisms of non-human animals.

## 1. Introduction

Reinforcement learning (RL) models have been widely used to simulate animal decision-making behavior [1,2]. The conventional RL model assumes that agents acquire action–outcome associations through trial and error. It is hypothesized that the option values are updated based on reward prediction error (RPE), which represent the disparity between actual and expected rewards [3]. These theoretical assumptions suggest that predictions of option values are gradually adjusted to reduce errors, and the magnitude of this adjustment is determined by the learning rate parameter.

Asymmetric learning rates have been studied in the context of reinforcement learning [4]. From the perspective of RL, learning rate bias refers to the difference in learning rates used for positive reward prediction errors and negative reward prediction errors. These learning rates refer to the difference in how animals learn from positive and negative reward prediction errors during the learning process. Research has shown that animals learn differently from positive and negative prediction errors, indicating an asymmetry in learning rates [5,6]. The phenomenon where the positive learning rate exceeds the negative learning rate is referred to as positive learning rate bias, while the opposite is known as negative learning rate bias. Positive learning rate bias indicates a greater emphasis on positive prediction errors, reflecting an enhanced learning response to unexpected rewards or better-than-expected outcomes. Conversely, negative learning rate bias refers to a greater sensitivity to negative prediction errors, highlighting a heightened response to unexpected punishments or worse-than-expected outcomes. Both positive and negative learning rate biases play significant roles in the adaptive learning and decision-making behavior of animals. These potential learning rate biases can be explained by risk-seeking/aversion behavior [7] or cognitive bias [8]. Previous studies have examined the asymmetry of learning rates across species [5,9], task contexts [10,11], and outcome types [6]. Many studies suggest that asymmetric learning rates tend to be biased in the direction of learning from positive RPEs, compared with negative RPEs [4,12]. Other studies indicate that negative learning rates are higher than positive learning rates [7,9]. However, all of these studies generally assume that the asymmetry remains stable throughout the learning process. How learning rate biases dynamically adjust as the learning process progresses remains to be investigated further.

Currently, most of the evidence regarding learning rate biases comes from behavioral data in humans and other animals. The primary role of learning rate biases is to modulate values through RPEs. Furthermore, previous studies have indicated that dopamine neurons in the midbrain exhibit similar firing patterns to both positive and negative RPEs [13,14,15]. These dopaminergic neurons project to the striatum of the basal ganglia, which is believed to play an important role in value encoding [16,17,18]. Previous studies have demonstrated the involvement of the striatum in representing value [19,20] and prediction errors [18,21]. Furthermore, as there are two types of dopamine receptor neurons involved in reward-based learning and avoidance learning, respectively, in the striatum, the asymmetry of the learning rate may stem from different responses to positive and negative rewards [22,23,24]. Therefore, we explored the dynamic patterns of learning rate biases from behavioral models to find signals related to the modulation of value by dynamic learning rate biases in the striatum.

Here, we recorded the behavioral data and the local field potentials (LFPs) in the striatum of five pigeons performing a free-choice probability task. Using a behavioral modeling approach, we dynamically analyzed the pigeons’ choice behavior and learning rate biases. We predicted that the pigeons’ learning rate biases dynamically transitioned from negative to positive along with the learning, during which the magnitude and valence of learning rate biases both changed. We hypothesized the striatal LFP power correlated with the option values of the pigeons, and the corresponding comparative and dynamic analyses in specific frequency bands were carried out.

## 2. Materials and Methods

### 2.1. Animals, Experimental Procedure and Surgery

Five adult and unsexed homing pigeons (*Columba livia*, numbered P014, P021, P029, P090, and P093) weighing 400 to 550 g were used in this study. All pigeons were individually housed in wire-mesh home cages (64 × 42 × 52 cm) under diurnal conditions (light for 12 h, dark for 12 h), deprived of food at 85% of their free-feeding body weight, and given free water. All experimental procedures involving animal experiments have been approved by the Life Science Ethics Review Committee of Zhengzhou University.

We designed a probability-learning task for pigeons, where they were free to choose between two options (Figure 1a). In each trial, rewards were provided according to predetermined probabilities, with one option corresponding to a higher rewarding chance than the other. The food supply for the pigeons was controlled to maintain their hunger state while ensuring an adequate water supply before the experiment. A closed experimental chamber measuring 50 × 50 × 50 cm was used for the pigeons, with a screen positioned in front of them presenting visual stimuli, as shown in Figure 1b. Below the screen were two keys corresponding to the stimuli that pigeons pecked to indicate their choice. To eliminate any potential bias, the positions of the visual stimuli were randomly swapped in each trial. Each trial began with a 5-s inter-trial interval (ITI), during which time the screen remained gray. This was followed by a 2-s stimulus presentation period, where green triangles (S+) and red squares (S−) simultaneously appeared on opposite sides of the central screen. During these 2 s, the pigeons were required to select one presented stimulus and confirmed their choice by pecking the corresponding key. Subsequently, the pigeons received a 3-s food reward with an 80% chance of receiving food when choosing the key associated with S+ and a 20% chance when selecting S− based on pre-determined reward probabilities. All trials were considered valid regardless of whether pigeons received rewards after pecking the keys. If no key pecking occurred within 2 s after the stimulus, it was considered an invalid trial and was followed by another ITI phase without any reward. Each session consisted of 50 trials, with a 10-min break between consecutive sessions. All pigeons completed two training sessions per day.

All animals were general anesthetized by intramuscular injection of 3% sodium pentobarbital (0.14 mL/100 g body weight), and then underwent microelectrode array implantation surgery. The pigeon’s head was fixed on a specially customized stereotaxic device, and then a small craniotomy was performed. According to the pigeon brain atlas [25,26], a 16-channel (4 × 4) microelectrode array was chronically implanted in the right striatal region (Figure 1c; AP 10.50 mm; ML ± 1.00 mm; DV 7.50 mm).

### 2.2. Data Acquisition

After a one-week postoperative recovery period, the pigeons began performing the free-choice experiment, in which the LFPs and behavioral data were simultaneously recorded. A multi-channel acquisition system (Cerebus^TM^, 128 channels, Blackrock Microsystems, Salt Lake City, UT, USA) was used for the amplification and collection of neural data. The raw neural signals were then filtered using a low-pass Butterworth filter with a cutoff frequency of 0–250 Hz to extract the LFPs.

### 2.3. Behavioral Learning Model

To quantitatively study the learning rate bias and the subjective option value of pigeons in probabilistic learning tasks, we used two models to fit the behavior data: Model 1 was a RL model with no learning rate bias (RLNB) [3,27], and Model 2 was a RL model with learning rate bias (RLB) [5,28,29,30].

In our experiment, although the rewards between options are of equal magnitude, their probabilities differ. Therefore, each trial assigns a reward value of 1 to the obtained reward and a value of 0 to the outcome without any reward:(1)$$r_{t}=\left\{\begin{array}{c} 1\;\;\;\;\;\mathrm{if}\;\mathrm{the}\;\mathrm{trial}\;\mathrm{is}\;\mathrm{rewarded}\\ 0\;\;\;\;\;\mathrm{otherwise} \end{array}\right.$$

At the start of each probabilistic choice task, the initial option’s values Q were set to 0, indicating an equal prior expectation of winning or not winning. After each trial, we updated the value of the selected option based on RPEs. RPE refers to the difference between the actual reward outcome and the expected outcome of the chosen option:(2)$$\delta_{t}=r_{t}-Q_{t}(a_{t})$$
where $a_{t}=S+(or\;S-)$ represents the choice of the S+ (or S−) at trial t, and $Q_{t}(S+)$ (or $Q_{t}(S-)$) represents the subjective value of S+ (or S−) at trial t.

Model 1 is a standard RL model without learning rate bias (also known as Q-learning), which updates the option value Q using the same learning rate for both positive and negative RPE:(3)$$Q_{t+1}\left(a_{t}\right)=\left.\begin{array}{c} Q_{t}\left(a_{t}\right)+\alpha \delta_{t} \end{array}\right.$$
(4)$$Q_{t+1}\left(a\neq a_{t}\right)=Q_{t}\left(a\neq a_{t}\right)$$
where α∈[0,1] represents the learning rate. Due to the adoption of a learning rate, there is no learning bias when updating values with positive or negative RPE.

Model 2 is a RL model with learning rate bias using two different learning rates $\alpha^{+}$ for positive RPEs ($\delta_{t}\geq 0$) and $\alpha^{-}$ for negative RPEs:(5)$$Q_{t+1}\left(a_{t}\right)=\left\{\begin{array}{c} {Q_{t}\left(a_{t}\right)+\alpha }^{+}\delta_{t}\;\;\;\;\;\;\;\mathrm{if}\;\delta_{t}\geq 0\\ Q_{t}\left(a_{t}\right)+\alpha^{-}\delta_{t}\;\;\;\;\;\;\;\mathrm{if}\;\delta_{t}<0 \end{array}\right.$$
(6)$$Q_{t+1}\left(a\neq a_{t}\right)=Q_{t}\left(a\neq a_{t}\right)$$
(7)$$Learning\;rate\;bias=\alpha^{+}-\alpha^{-}$$

Due to the adoption of the dual learning rate, there is learning rate bias when updating values with positive or negative RPE. When $\alpha^{+}>\alpha^{-}$, value learning exhibits a positive learning rate bias, meaning it places more emphasis on positive outcomes. When $\alpha^{+}<\alpha^{-}$, value learning exhibits a negative learning rate bias, meaning it places more emphasis on negative outcomes.

Both models use the SoftMax rule to convert the value of options into the probabilities of selecting those options:(8)$$P_{t}\left(a_{t+1}=i\right)=\frac{\exp\left(\beta Q_{t}\left(i\right)\right)}{\sum_{i}\exp\left(\beta Q_{t}\left(i\right)\right)}$$
where *β* represents the inverse temperature parameter, determining the degree of sensitivity in the selection probability to variations in Q values. As it increases, the probability of choosing a certain option becomes more sensitive to the difference in value between two options.

To investigate the variations of positive and negative learning rates at different learning stages of pigeons, we initially partitioned the behavioral data of each pigeon into 10 overlapping windows representing 10 learning stages. We performed parameter recovery for the two models using simulated behavioral choice data containing different numbers of trials. To ensure a parameter recovery rate of over 85%, we ultimately determined that each window contained 200 trials. Subsequently, separate positive and negative learning rates, along with inverse temperature parameters for each window, were fitted. 

The method of minimizing the negative logarithm of posterior probability (*NLPP*) is used for parameter optimization [31]:(9)$$NLPP=-\log\left(P\left(\theta_{M}|D,M\right)\right)\propto -\log\left(P\left(D|M,\theta_{M}\right)\right)-\log\left(P\left(\theta_{M}|M\right)\right)$$
where $P\left(D|M,\theta_{M}\right)$ is the likelihood of the behavior data *D* (i.e., choice sequence) given the current model (i.e., Model 1 or Model 2) and its parameters $\theta_{M}$ ($\theta_{M1}=(\alpha ,\beta )$, $\theta_{M2}=(\alpha^{+},\alpha^{-},\beta )$), and $P\left(\theta_{M}|M\right)$ is the prior probability of the parameters.

We choose Beta(1.1, 1.1) as the prior distribution for the learning rate and Gamma(1.2, 5) as the prior distribution for the inverse temperature parameter to prevent extreme parameter estimates while ensuring a broad prior [31]. The parameter search was initialized by randomly selecting a starting point within certain ranges and implemented using Matlab’s *fmincon* function.

### 2.4. LFP Data Analysis

Before analysis, we downsampled the raw LFP signals to 1000 Hz. Subsequently, we performed baseline drift removal and eliminated 50 Hz power interference using a notch filter. To extract the power of multiple frequency bands in a single trial, the LFP signals were first band-pass filtered, and Theta (5 to 8 Hz), Alpha (9 to 12 Hz), Beta (13 to 30 Hz) and Gamma (31 to 80 Hz) band signals were obtained. The power spectrum was calculated during the 0.5 s period after the decision was made using the Morlet wavelet method, which corresponded to the period of the reward outcome of the pigeon. We computed the average power of all trials within four distinct frequency bands separately. Trials where LFP power exceeded five times the average power of that specific band were excluded. To avoid false results due to the temporal nature of data during the whole learning process, we normalized the power in each band during outcome periods by dividing the baseline period (ITI) power over 0.5 s for each trial.

The LFP power of all bands during the early and late learning stages were analyzed to confirm the most relevant frequency band for learning. Subsequently, we calculated the Spearman correlation coefficients between LFP power and S+ values, S− values, and RPEs estimated by both Model 1 and Model 2 trial-by-trial. Then, we established six linear regression models to further study the correlation between the LFP power and variables estimated by RL models. The dependent variable in all six linear regression models is the Gamma-band power, while the independent variables are S+ values, S− values and RPEs, respectively, estimated by both Model 1 and Model 2.

### 2.5. Statistical Analysis

We employed the Shapiro–Wilk test to assess the normality of the data and determine whether parametric or non-parametric tests should be used. An unpaired t-test was used to evaluate statistical differences for normally distributed data, while the Wilcoxon signed-rank test was used for non-normally distributed data. The results are presented as mean ± standard error with a significance level set at 5%. In the figures, * represents *p* < 0.05, ** represents *p* < 0.01, *** represents *p* < 0.001, and **** represents *p* < 0.0001. Statistical analyses were performed using Matlab 2022b, GraphPad Prism 10.0.2, and OriginPro 2023 Student Edition software.

## 3. Results

We analyze the evolution of the pigeon’s value and RPEs during the learning process, as well as the correlation between the striatal LFP signals during the outcome phase and the learning variables estimated by two RL models.

### 3.1. Behavioral Results

During the initial stages of learning, the accuracy of behavior for five pigeons is approximately 50%, indicating a random selection process. As learning progresses, pigeons increasingly favor the key associated S+, which offers a higher probability of reward. Eventually, the proportion of correct choice (selecting S+) stabilizes at around 95%, as depicted in Figure 2a. The evolution of behavioral performance suggests that pigeons learn through a process of trial-and-error.

To dynamically quantify the learning rate biases of pigeons in the 10 stages of value learning according to chronological order, two RL models were used to fit the behavioral data. The results show that both models effectively simulated the behavior (Figure 2b). Importantly, the fitted positive and negative learning rates of Model 2 did not remain consistent throughout the learning progress (Figure 2c). A larger negative learning rate ($\alpha^{-}$) than the positive learning rate ($\alpha^{+}$) was observed in the early learning stages, indicating that pigeons quickly learnt from the negative events without rewards compared with the positive events. Conversely, a larger positive learning rate was observed in the late stages. The results suggest that once pigeons had acquired some knowledge about option values, they primarily learnt from the correct-choice events, namely positive events associated with receiving rewards. In addition, the inverse temperature parameter β increased as the learning progressed (Figure 2d, R^2^ = 0.7265, ** *p* = 0.0017), indicating that subjects’ random exploration gradually decreased with learning progress, i.e., the choice probability based on option values increased. The learning rate bias showed a linear increase with the learning stage (Figure 2g, R^2^ = 0.6545, ** *p* = 0.0046). In the early stages there was a negative bias, which later shifted to a positive bias. The correlation between learning rate bias and choice accuracy (Figure 2e, R^2^ = 0.6086, ** *p* = 0.0078) was higher than that with the inverse temperature parameter (Figure 2f, R^2^ = 0.4324, * *p* = 0.0382). The model parameter results of other pigeons are shown in Appendix A. These results indicate a significant correlation between the variation in pigeon dynamic learning rate bias and behavioral performance. In other words, during the early learning stages, the learning rate bias is negative when the accuracy is low, while the learning rate bias linearly increases to positive values as the accuracy improves. Option values and RPE estimation results show significant differences between option values and RPEs estimated by Model 1 and Model 2 (Figure 2h,i, V(S+), **** *p* < 0.0001, *t* = 6.83; V(S−), **** *p* < 0.0001, *t* = 18.97; RPEs, **** *p* < 0.0001, *t* = 11.06; paired two-tailed *t*-tests).

### 3.2. Choice Modulates Gamma LFP Power of Outcome Phase

The average power of four bands integrated over a time window of 0.5 s after the outcome presentation was computed to investigate the correlation between the striatal LFP activity and different options. We found that the power in all four bands during the late learning stage was significantly higher compared with the early stage when choosing S+ trials (Figure 3a, *p* < 0.0001). Moreover, the greatest difference in Gamma band was observed between the late and early stages when choosing S− trials (Figure 3a, * *p* = 0.0107 for Theta, ** *p* = 0.0082 for Alpha, * *p* = 0.0119 for Beta, ** *p* = 0.0030 for Gamma). Further, significant power differences were observed in all four bands between choosing S+ and S− trials during the late learning stages (Figure 3a, * *p* = 0.0134 for Theta, ** *p* = 0.0069 for Alpha, ** *p* = 0.0051 for Beta, ** *p* = 0.0032 for Gamma). Considering that the most significant differences lie in the Gamma band, we focused on Gamma to explore the dynamic power changes throughout the entire learning process. As shown in Figure 3b, the power gradually increased with each session and reached its peak during the late stages before slightly decreasing. The increased Gamma power was primarily driven by selecting S+ trials, while no significant increase was observed for selecting S− trials during the early stages. The power is significantly greater for trials of choosing S+ than for choosing S− trials during almost the entire learning process (Figure 3b). Figure 3c shows the changes of Gamma power between rewarded and unrewarded trials. It is observed that the power of rewarded trials is significantly higher than that of unrewarded trials during the early stages. However, this difference disappears in the late stages. Furthermore, there is almost no significant difference in power between rewarded and unrewarded trials of choosing S+ (Figure 3d). These results show that Gamma power modifications in the striatum of pigeons are differentially modulated by the choice, and may be involved in representing option values during learning processes.

### 3.3. Gamma-Band LFP Correlates of Values Modulated by the Dynamic Learning Rate Bias

To investigate the relationship between Gamma power in the striatum and the option values and RPEs during the outcome period, we computed the Spearman correlation between them on a trial-by-trial basis. The results revealed a high correlation between the Gamma power and the estimated option values from both models, while the correlation with RPE error was low (Table 1). Four linear regression models were constructed for further correlation analyses between Gamma power and estimated values (Figure 4). The results demonstrated the strongest correlation between Gamma power and the estimated value of option S+ in Model 2 (Figure 4b, R^2^ = 0.6656, **** *p* < 0.0001).

## 4. Discussion

Our results show that in the free-choice probabilistic learning task, the learning rate bias of pigeons linearly increases during the learning process, gradually shifting from a negative bias to a positive bias. In the early learning stages, pigeons tend to prioritize updating the values of options from unrewarded outcomes to reduce risk. Conversely, they focus on updating values from positive outcomes, and further increasing their belief in making choices in the late stages. Additionally, our study indicates a correlation between pigeons’ striatal Gamma power and the value modulated by dynamic learning rate bias. Overall, our results suggest that pigeons use dynamic learning rate bias in the process of value learning from both behavioral and neural perspectives.

In the probabilistic learning task presented in this study, pigeons demonstrated a longer learning duration compared to humans [32] and other experimental animals such as monkeys [18] and rats [33]. That provides the opportunity to dynamically investigate variations in pigeons’ learning rate bias throughout the entire value-learning process in the context of reinforcement learning. Previous extensive studies have shown that animals differ in the extent to which they learn from positive and negative prediction errors, suggesting learning rate bias [5,6,7,8,9,12]. When $\alpha^{+}<\alpha^{-}$, the influence of negative prediction errors outweighs positive ones on value learning, resulting in behaviors indicative of risk aversion [7]. In the initial stages of learning, we observed that pigeons displayed a prudent strategy, updating their option values based on unrewarded, negative outcomes as a means to circumvent risk. This cautious approach can be construed as a cognitive strategy that prioritizes survival instincts amidst uncertain conditions. This risk-averse behavior gradually decreased as the learning process progressed, and vanished when pigeons learnt to choose the optimal option. After the pigeons learnt to select advantageous options, they began to adopt a more optimistic learning rate bias.

The observed linear increase in the learning rate bias, transitioning from negative to positive, is suggestive of an adaptive learning mechanism. During the early stages, the pigeons exhibit a conservative approach, preferring to adjust expectations based on non-rewards to mitigate initial risk. This conservative bias potentially represents a strategic allocation of attention towards avoiding losses when faced with uncertainty [34,35]. As proficiency within the task grows, the evolution of a positive learning bias embodies a shift in strategy—pigeons start to favor learning from positive reinforcement, bolstering their confidence in decision-making processes. This progression from caution to confidence may reflect an optimization of learning, where the adaptation and retention of advantageous behaviors are prioritized as the familiarity with contingencies improves [36]. Learning rate bias is also influenced by the scarcity of rewards in the environment [5]. In environments with scarce rewards, positive learning rate biases can cause animals to overlook failures and exploit rare rewards. Therefore, the dynamic adjustment of learning rate biases may reflect adaptive behavior in animals. Furthermore, with the learning process, the inverse temperature parameter gradually increases, which also reflects the adaptive learning behavior of pigeons in terms of exploration and exploitation. The inverse temperature parameter reflects the degree of exploration in the learning process of pigeons. The larger it is, the more certain the animals’ choices are made based on the value of options, and the lower the degree of random exploration.

The evolutionary trajectory of option value estimates, as yielded by our behavioral models, presents a significantly divergent pattern in the presence versus absence of learning rate biases. The striatum has long been recognized as a hub of reward-related information processing, with its role in the value encoding [37,38,39,40] and prediction errors [18] being well documented. The Gamma-band oscillations in the striatum are thought to represent critical computational processes related to the integration and coordination of neural activity underlying decision-making [41]. Our results indicate that the pigeons’ striatal Gamma powers are correlated with the option values modulated by dynamic learning rate bias, which neurologically demonstrates that pigeons may adopt dynamic learning rate bias in learning tasks.

Taken together, our findings support that pigeons adopt a dynamic learning rate bias during their learning process, i.e., the valence of learning rate bias shifts from negative to positive as learning progresses. Moreover, our results suggest that the pigeon’s striatum is involved in the value representation modulated by dynamic learning rates. These results help us to understand the fundamental learning processes shared by different species, providing valuable insights into the reinforcement learning mechanisms in non-human animals.

## Figures and Tables

**Figure 1 animals-14-00489-f001:**
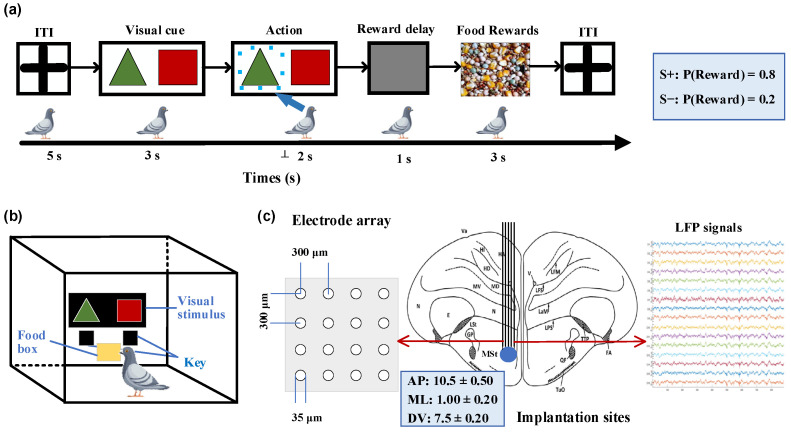
Probabilistic learning task, experimental setup, and implantation sites. (**a**) Sequence of events in a single trial. After an inter-trial interval (ITI), a pair of visual stimuli with different colors and shapes were displayed on the screen. The pigeon was required to peck one of two keys corresponding to the stimuli within a 2-s timeframe. After a 1-s delay, the pigeon could receive a food reward for 3 s, determined by specified reward probabilities. Pecking the key associated with S+ (green triangle stimulus) corresponds to an 80% rewarding chance, while pecking S− (red square stimulus) corresponds to 20%. (**b**) Experimental setup. The experimental setup was a rectangular chamber with dimensions of 50 cm in length, width, and height. One side of the chamber had a door that can be opened. The chamber included a screen (80 × 60 mm) on the wall opposite to the door for the display of visual stimuli, two keys corresponding to the stimuli on each side below the screen, and a food tray (40 × 30 mm) for rewarding food. (**c**) Diagram illustrating the implantation site of microelectrode array in the medial striatum and example local field potential (LFP) signals. AP: anteroposterior, ML: mediolateral, DV: dorsoventral.

**Figure 2 animals-14-00489-f002:**
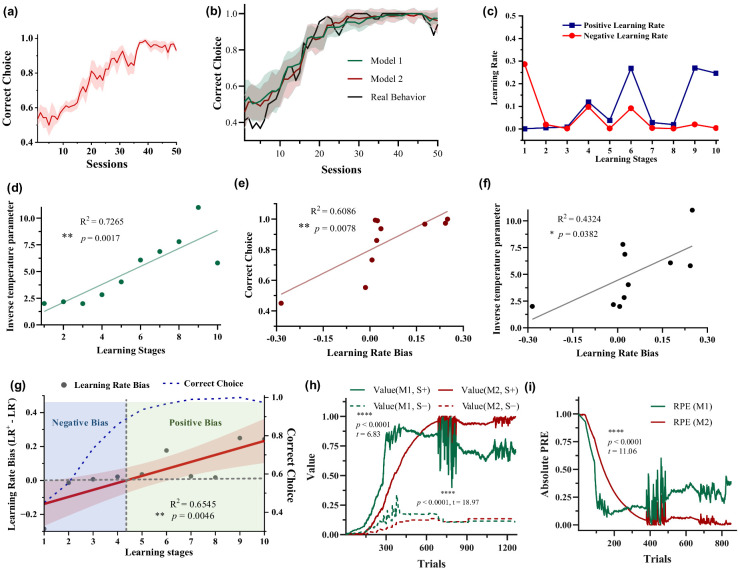
Behavioral performance and the results of the two RL models. (**a**) Choice performance of five pigeons. The solid line and shaded area correspond to the mean ± standard error of the mean (SEM) of the probability of choosing the most rewarding stimulus S+ across sessions. (**b**) Comparison between choice performance computed by two RL models and actual choice performance. The solid line and shaded area correspond to the mean ± standard deviation (SD) of the probability of choosing the most rewarding stimulus S+ across sessions simulated by models. (**c**,**d**) Dynamic learning rate and inverse temperature parameter β of Model 2 (RLB) during the learning process (P090 as an example). (**e**,**f**) Relationship between learning rate bias and choice performance, inverse temperature parameter β. (**g**) Dynamic learning rate (LR) bias and choice performance as a function of learning stages. (**h**,**i**) Option values and RPEs estimated by Model 1 and Model 2.

**Figure 3 animals-14-00489-f003:**
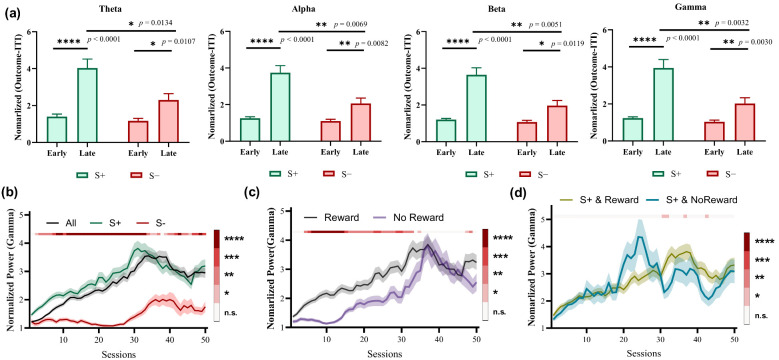
LFP power of two choices in the striatum during the outcome phase (n = 5, for each animal, number of early-stage trials = 100, number of late-stage trials = 100). (**a**) LFP powers at different bands in the early and late learning stages. (**b**) The dynamic changes of Gamma power grouped by choice (All, S+, or S−). The color bar shows the statistical test results between S+ and S− choices. (**c**) The dynamic changes of Gamma power grouped by outcome (Reward, No Reward). The color bar shows the statistical test results between Reward and No Reward outcomes. (**d**) The dynamic changes of Gamma power of trials choosing S+ grouped by choice and outcome (S+ & Reward, S+ & No Reward). The color bar shows the statistical test results between S+ & Reward and S+ & No Reward. (**b**–**d**) * *p* < 0.05, ** *p* < 0.01, *** *p* < 0.001, and **** *p* < 0.0001.

**Figure 4 animals-14-00489-f004:**
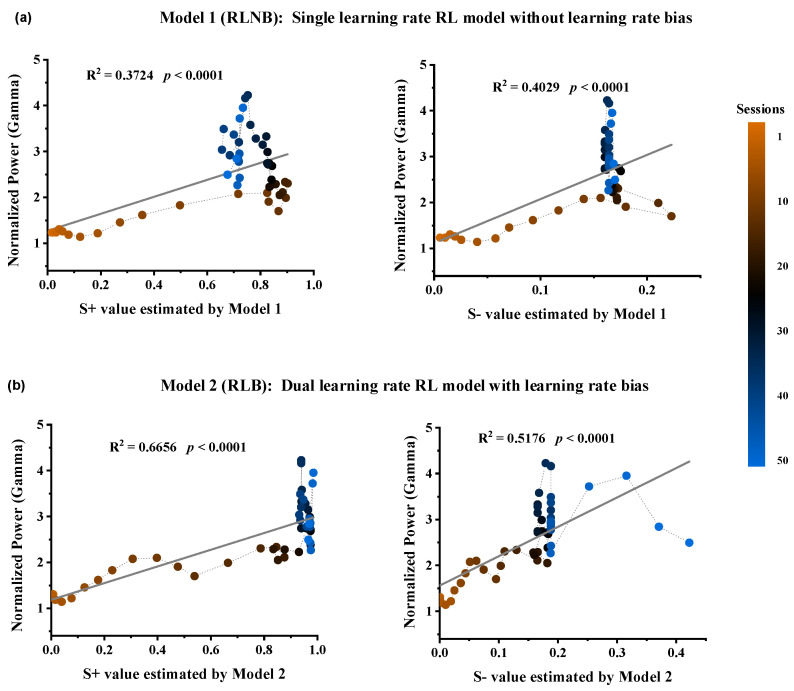
Relationship between Gamma power and the estimated option values from the two RL models. (**a**) Option S+ values (DF = 48, R^2^ = 0.3724, F = 28.488, *p* < 0.0001) and Option S− values (DF = 48, R^2^ = 0.4029, F = 32.434, *p* < 0.0001) estimated by Model 1. (**b**) Option S+ values (DF = 48, R^2^ = 0.6656, F = 95.531, *p* < 0.0001) and Option S− values (DF = 48, R^2^ = 0.5176, F = 51.501, *p* < 0.0001) estimated by Model 2.

**Table 1 animals-14-00489-t001:** Spearman correlation between Gamma power in the outcome period and the variables estimated by Model 1 and Model 2.

	S+ Value (Model 1)	S− Value (Model 1)	S+ Value (Model 2)	S− Value (Model 2)	RPE (Model 1)	RPE (Model 2)
Spearman correlation	0.2303	0.1747	0.3647	0.3309	0.0579	0.0093

## Data Availability

The datasets analyzed in the current study are available from the corresponding author upon reasonable request.

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
