# Peer review of "Dynamics Learning Rate Bias in Pigeons: Insights from Reinforcement Learning and Neural Correlates"

_animals, 2024, doi:10.3390/ani14030489_

Round 1

Reviewer 1 Report

Comments and Suggestions for Authors
  1. Whereas previous analyses of animal behavior were conducted under the assumption that learning rates and inverse temperatures were constant, this paper reveals that they change as training progresses.

  2.  
  3. Major Issue:

  4. Inaccurate Citation of References: The paper suffers from incorrect citation of references, affecting its credibility. References appear shuffled and do not correspond to the text. For example, references [6-8], which are about humans, are incorrectly cited for animal studies. References [12, 15], which are relevant to animal studies, should have been used instead. Reference [5], cited in P2, L51, does not deal with asymmetry, and [16], though theoretical, is inaccurately cited as experimental. The paper should better utilize references [12] and [16] in its discussion, especially regarding the impact of asymmetry on survival strategies. References [6][7] suggest pseudo-effects in asymmetry, and if cited, the paper should incorporate their proposed mechanisms beyond just learning rates.

  5.  
  6. Minor Issue:

  7. Lack of Individual Variation and Statistical Data in Fig.2(c-g): The paper fails to present individual differences and statistical information regarding the learning rates and inverse temperature of the subjects. It is unclear whether these were averaged across individuals or not.

  8.  
  9. Suggestion for Discussion: The paper should discuss the correlation between learning rate bias and inverse temperature, as well as the time evolution of inverse temperature.

  10.  

Reviewer 2 Report

Comments and Suggestions for Authors

In this study, the authors conducted a comprehensive investigation into reinforcement learning mechanisms in pigeons through the analysis of behavior data and local field potentials (LFPs) recorded in the striatum. They found that pigeons tend to learn more from the negative outcomes initially, and they start to learn more from the positive outcomes as the learning progresses. This study presents valuable insights into the dynamic learning strategies employed by non-human animals during a probabilistic learning task. I have only minor comments and I recommend acceptance pending these revisions.:

1. The manuscript mentions the use of reinforcement learning models, but it would be helpful to specify which specific RL algorithm was employed. I’m not sure what is the standard (Line 164) RL model. Are the authors using the Q-learning algorithm?

2. The authors discuss positive and negative reward prediction errors and learning rate biases. Providing a brief explanation or citation on how positive and negative learning rates are calculated in the context of reinforcement learning models would aid readers' understanding.

3. Clarify the definition of a "correct choice." Is it defined as selecting S+ or the choice leading to the receipt of food?

Comments on the Quality of English Language

Overall, the English is good, but there are minor errors:

Line 15: "We found that 15 pigeons shown a tendency..." → Suggested revision: " We found that 15 pigeons showed a tendency..."

Line 154: "In our experimen,..." → Suggested revision: "In our experiment,..."
